# ConvART: Improving Adaptive Resonance Theory for Unsupervised Image Clustering

**Ilia Sucholutsky & Matthias Schonlau**
Department of Statistics and Actuarial Science
University of Waterloo
Waterloo, ON N2L3G1, Canada
{isucholu,schonlau}@uwaterloo.ca

## Abstract

While supervised learning techniques have become increasingly adept at separating images into different classes, these techniques require large amounts of labelled data which may not always be available. We propose a novel method for unsupervised image clustering by combining Adaptive Resonance Theory (ART) with techniques from Convolutional Neural Networks (CNN). ART networks are unsupervised clustering algorithms that have high stability in preserving learned information while quickly learning new information. Meanwhile, a major property of CNNs is their translation and distortion invariance, which has led to their success in the domain of vision problems. By embedding convolutional layers into an ART network, the useful properties of both networks can be leveraged to identify different clusters within unlabelled image datasets and classify images into these clusters. In exploratory experiments, we demonstrate that this method greatly increases the performance of unsupervised ART networks on a benchmark image dataset.

## 1 Introduction

Supervised deep learning techniques are achieving increasingly impressive results on a wide range of vision problems (Mathieu et al., 2015; Provodin et al., 2016; Zhang et al., 2017). In particular, supervised learning techniques have been shown to be able to separate images into classes with incredible accuracy, even surpassing human performance (Schmidhuber, 2015). Convolution Neural Networks (CNN) have been one of the major drivers of this progress in solving vision problems. Since LeCun et al. (1998) first described CNNs, numerous improvements have been made to them (Krizhevsky et al., 2012; Simonyan & Zisserman, 2014; Girshick et al., 2014).

However, supervised learning depends heavily on the availability of labelled data. Unsupervised learning techniques aim to solve this problem by working entirely independently of data labels. ART was originally proposed as a solution to the plasticity-stability dilemma of quickly learning new knowledge without disrupting what was already learned (Grossberg, 1987), and led to the development of several unsupervised techniques under the neural network framework (Carpenter & Grossberg, 1987a;b; 1990; Carpenter et al., 1991b).

We aim to leverage the benefits of both network types by embedding convolutional layers into ART networks in order to create a novel unsupervised method for discovering classes in image datasets.

## 2 Theory

### 2.1 ART

The idea behind ART is to have a self-organizing network that creates a new template for every class it identifies within data. As the network observes new data, it compares it to the templates it has already learned. If any of the templates are sufficiently similar to the input pattern (as dictated by a vigilance parameter), then the input is assigned to the class of the best-matching template, and the

template is updated to include this input. If none of the templates sufficiently match the input, then a new class is created and the input is used to create the template for this new class. A typical ART network has 3 main components: The 'F1(a)' layer is an input layer with one node for each input dimension. The 'F1(b)' layer is the the comparison layer where the level of match between input pattern and template pattern is determined. It has one node for every node in 'F1(a)'. The 'F2' layer is self-organizing and has one node for each detected class. 'F1(b)' and 'F2' are bidirectionally fully-connected. When a new input arrives at 'F1(a)' it is sent to 'F1(b)'. The signal is then sent on to 'F2' multiplied by the 'bottom-up' weights. The resulting values produced at each 'F2' node are used to efficiently search for a matching class going from highest to lowest. To see if a certain class matches the input, signal multiplied by the 'top-down' weights is sent back from the 'F2' node being considered to the 'F1(b)' layer. Within the 'F1(b)' layer the incoming input pattern is compared to the incoming template pattern using a match function that varies between implementations of ART. Regardless, if the output of this function is higher than the vigilance parameter, the input is considered to match the class, and learning takes place where the top-down and bottom-up weights are updated. Otherwise the next node with highest output is considered, or if none are left, then a new class is created and the input assigned to it. It has been shown that the algorithmic behaviour of many of the ART implementations can be fully described using a set of competitive differential equations (Carpenter et al., 1991a).

## 2.2 CNN

One of the issues when working with image data is the large number of inputs. Even small images like those in CIFAR-10 are 32 pixels high, by 32 pixels wide, with 3 color channels, meaning 3072 inputs for every image (Krizhevsky & Hinton, 2009). Clearly, simply representing an image by a vector of the values for each pixel feeding this to a fully-connected network is not a scalable solution. However, even when working with small images, there arises a second issue with a simple vector representation. Images often have internal spatial relationships but vary due to translation, scaling, and distortion. For example, the digit '8' has two similarly-sized, stacked circles, but as long as they are together one on top of the other, these 2 circles can occur anywhere in the image and one would still have an image of the digit '8'. When the image matrix is simply represented as a vector in a fully-connected network, a lot of this spatial information is lost. ART networks struggle with these 2 problems as large inputs lead to large template patterns, while small translations within an image can lead to an input pattern appearing significantly different from the existing template.

However, CNNs repeatedly make use of 2 special layers to solve these issues: Convolutional layers can be thought of as passing a small filter over the entire image in a windowed way. Each neuron receives only the result of applying this same filter to one particular window of the image. This provides varying degrees of translation, scale, and distortion invariance while preserving spatial relations as the same transformation is performed on each part of the image. Pooling layers essentially perform down-sampling by providing some sort of smaller descriptive statistic about each window. For example, max pooling outputs the maximum value of each window. This reduces the dimension of the data. Of course, other layers and elements are also present within CNNs: Rectified Linear Units (ReLU) perform non-linear transformations of the data, and fully-connected layers are used in the typical way.

## 2.3 EMBEDDING CONVOLUTIONS INTO ART

While the windowed filtering analogy is useful for conceptually understanding the operation of convolutional layers, the reality is that these layers are simply sparse, weight-sharing counterparts of fully-connected layers. As such, if filter dimensions are pre-determined, it is straightforward to embed a block of convolutional layers into an ART network. The convolutional block simply takes in and transforms input from the 'F1(a)' layer of the ART network. The output from the block is then treated the same way that output from the 'F1(a)' layer of an ART network is treated and sent to the 'F1(b)' layer. In such a way, we extend the 'F1(a)' layer of the ART network to perform a more complex transformation and encoding of the input pattern. It is important that our modification be an extension of specifically the 'F1(a)' layer as this preserves the differential equations governing the ART network learning dynamics which occur exclusively between 'F1(b)' and 'F2'. The resulting ConvART network still provides a solution to the plasticity-stability dilemma while taking advantage of the translation and distortion invariance properties of CNNs to extend ART to the image domain.

## 3    EXPERIMENTAL RESULTS

We used normalized MNIST data for our experiment (LeCun & Cortes, 2010). We used our proposed method to embed an ART1 network as described by Carpenter & Grossberg (1987a), with 4 additional layers: a convolutional layer with all 512 possible binary 3-by-3 filters pre-defined and ReLU activation, a max pooling layer reducing the dimension by a factor of 3, another convolutional layer with all 16 possible binary 2-by-2 filters pre-defined and ReLU activation, and finally, a max pooling layer reducing dimension by a factor of 3. The final outputs of the model are 16 2-by-2 matrices for each image, which we reshaped into a vector of length 64.

We randomly selected 350 MNIST images. The number of samples of each digit can be seen in the last column of Table 1 in the Appendix. Since we extended an ART1 network which works best with binary data, we first pre-processed the normalized MNIST data by rounding all the values to the nearest integer. We also normalized the output of the final pooling layer and rounded these normalized values.We stream the samples to our ConvART network with a decaying vigilance parameter with initial value of 0.75, minimal value of 0.45, and a decay rate of 1.01. Since the data is streamed in, at each step the model both classifies and learns.

The proposed model detected 40 classes. To understand performance, we recorded a matrix of how many times each number was assigned to each class as visualized in Figure 1. While the number of predicted classes is high, Figure 2 shows each digit had less than 5 classes to which it was mostly assigned. Meanwhile, Figure 3 shows that each large class corresponded mainly to just one digit, although it is clear that the network struggled to separate certain pairs of numbers. For example, examining class 25 in Figure 1 reveals that the network had trouble separating the numbers 1 and 4.

Running an ART1 network with the same vigilance parameter and decay on the same 350 pre-processed images leads to detection of 334 classes, 8.35 times as many as our proposed model, suggesting ART1 did not recognize similarities between most images and placed them in separate classes. To reduce this to 40 classes, the vigilance parameter was set to a constant 0.016. Figure 4 visualizes the classification matrix for this model sorted on the y-axis. Digits were assigned to many more classes, and individual classes were assigned a larger quantity of different digits, than when using our ConvART model. Our proposed model led to a 44% increase in the quantity of digits found within their respective 3 top classes. More detail on the proportion of digits classified within their respective largest 3 classes can be found in Table 1 in the Appendix.

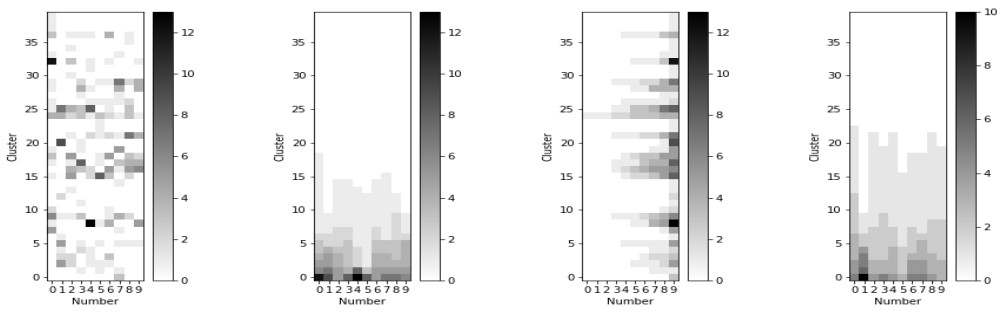

Figure 1: ConvART un-sorted class/digit matrix    Figure 2: ConvART y-sorted class/digit matrix    Figure 3: ConvART x-sorted class/digit matrix    Figure 4:    ART1 y-sorted class/digit matrix

## 4    CONCLUSION

We have demonstrated that extending ART networks with convolutional components results in a novel technique for unsupervised image clustering. The proposed method led to a 44% increase in the number of MNIST digits classified into their top 3 classes relative to the ART1 model when the models were configured to return the same number of classes, and an over 8-fold decrease in the number of predicted classes when parameters were held constant across the two models.

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

## A    FUTURE WORK

It is clear that there are several areas of improvement for this method.

First, the performance of our method while better than regular ART1, is still inferior to classical techniques like k-means clustering. In large part, we expect that this is due to the streaming nature of our algorithm as compared to k-means which trains on the entire dataset at once. However, it is also likely that performance was impacted by the use of ART1 rather than other more complex ART networks. The first extension to this work will be experimenting with embedding convolutions into ART2 and FuzzyART to see if performance improves.

Second, one of the issues affecting most ART implementations as well as numerous other clustering algorithms, is the lack of a mechanism for merging clusters, which leads to a large number of clusters containing a small number of points each. We are currently developing a mechanism by which at any point during training, if two clusters are found to be sufficiently similar, they will be merged into one.

Third, in this small implementation, we generated all possible binary 3-by-3 and 2-by-2 filters. However this method is not scalable as generating all filters of larger sizes or non-binary filters would be at least exponentially more difficult. Two directions are plausible here, using a high-number randomly generated filters, or learning the filters incrementally as the network trains.

Nonetheless, we have shown that by embedding convolutions into ART networks, we can extend a powerful unsupervised clustering algorithm to also work on image data.

## B    ADDITIONAL DETAILS

Table 1: Proportion of MNIST digits classified within their respective largest 3 classes

| Digit | ConvART1 Proportion | ART1 Proportion | Total # in sample |
|-------|---------------------|-----------------|-------------------|
| 0     | 0.489               | 0.277           | 47                |
| 1     | 0.568               | 0.568           | 37                |
| 2     | 0.412               | 0.294           | 34                |
| 3     | 0.429               | 0.314           | 35                |
| 4     | 0.611               | 0.278           | 36                |
| 5     | 0.545               | 0.318           | 22                |
| 6     | 0.4                 | 0.371           | 35                |
| 7     | 0.444               | 0.361           | 36                |
| 8     | 0.444               | 0.278           | 36                |
| 9     | 0.469               | 0.281           | 32                |
| Total | 0.483               | 0.334           | 350               |

