# OpenReview forum: "ConvART: Improving Adaptive Resonance Theory for Unsupervised Image Clustering"
_ICLR.cc/2018/Workshop — Reject_

### Official Review · AnonReviewer1 · 2018-03-14
**Clustering, other work, performance measures**

**Rating:** 4
**Confidence:** 4

**Review:**

The paper discusses clustering of MNIST digits using a combination of adaptive resonance theory (ART) and convolutional neural networks (CNNs). Improvements of the combined system are then shown relative to the ART system alone. Measures are number of found clusters and frequencies of assignments.

ART networks are typically tightly coupled to models of neural information processing; and it is important to investigate neuro-dynamic models for learning. Also neuro-scientifically motivated approaches frequently use MNIST for evaluation
(e.g.,Neftci et al different years, Nessler et al., PLOS CB 2013, Keck et al. PLOS CB 2012, Schmuker etal PNAS 2013 etc). Performance measures are classification (e.g. using few labels, or hand-assigned labels after clustering), also less neuroscientific approaches (Boltzmann Machines, SBNs etc) are frequently used and provide, e.g., likelihoods as performance measures. These approaches are not mentioned, and it is unclear how the reported performance should be interpreted in relation to these other works. Note that many of the above mentioned systems seem to have less problems in separating digits, so some mechanisms might be useful for the line of research pursued here.

In general, combinations of deep learning ideas and neuro-dynamic approaches are a good idea. Here, I feel that neither performance measures nor the discussion of the literature is (at least currently) sufficient (also given that this is a workshop paper).

---

### Official Review · AnonReviewer4 · 2018-03-17
**Motivation and context?**

**Rating:** 4
**Confidence:** 4

**Review:**

It's not clear what the motivation for this work is. The citations to ART are ~30 years old, and the method is far from working well as a current clustering method. If the target is a neuroscience audience, that isn't clear: more connection to their ongoing work is required. If the target is those interested in furthering the state of the art in representation learning, the idea needs to be further along.

---

### Decision · Program_Chairs · 2018-03-20
**ICLR 2018 Workshop Acceptance Decision**

**Decision:**

Reject

**Comment:**

Based on the reviews, this paper has not been accepted for presentation at the ICLR workshop. However, the conversation and updates can continue to appear here on OpenReview.